# ALFA: Adversarial Feature Augmentation for Enhanced Image Recognition

## Abstract

*Adversarial training* is an effective method to combat adversarial attacks in order to create robust neural networks. By using an auxiliary batch normalization on adversarial examples, it has been shown recently to possess great potential in improving the generalization ability of neural networks for image recognition as well. However, crafting pixel-level adversarial perturbations is computationally expensive. To address this issue, we propose **A**dversaria**L F**eature **A**ugmentation (ALFA), which advocates adversarial training on the intermediate layers of feature embeddings. ALFA utilizes both clean and adversarial augmented features jointly to enhance standard trained networks. To eliminate laborious tuning of key parameters such as locations and strength of feature augmentations, we further design a *learnable* adversarial feature augmentation (L-ALFA) framework to automatically adjust the perturbation magnitude of each perturbed feature. Extensive experiments demonstrate that our proposed ALFA and L-ALFA methods achieve significant and consistent generalization improvement over strong baselines on CIFAR-10, CIFAR-100 and ImageNet benchmarks across different backbone networks for image recognition.

## 1 Introduction

Neural networks often fall vulnerable when presented adversarial examples injected with imperceptible perturbations, and suffer significant performance drop when facing such attacks (Szegedy et al., 2013; Goodfellow et al., 2015b). Such susceptibility has motivated abundant studies on adversarial defense mechanisms for training robust neural networks (Schmidt et al., 2018; Sun et al., 2019; Nakkiran, 2019; Stutz et al., 2019; Raghunathan et al., 2019), among which *adversarial training* based methods (Madry et al., 2018b; Zhang et al., 2019a) have achieved consistently superior robustness than others.

The general focus of adversarial training is to enhance the robustness of gradient-based adversarial examples. A few recent studies (Zhu et al., 2020; Gan et al., 2020) turn to investigate the generalization ability of adversarial training on language models. However, in-depth exploration of extending this to the vision domain is still missing. Xie et al. (2020) proposes to utilize adversarial examples with an auxiliary batch normalization to improve standard accuracy for image recognition, but it still suffers from expensive computational cost from the generation of pixel-level perturbations.

To address this issue, we propose **A**dversaria**L F**eature **A**ugmentation (ALFA) as a natural extension of adversarial training, with a focus on leveraging adversarial perturbations in the *feature space* to improve image recognition on clean data. As illustrated in Figure 1, ALFA introduces adversarial perturbations to multiple intermediate layers. These perturbed feature embeddings act as a special feature augmentation and *implicit regularization* to enhance the generalization ability of deep neural networks. Consequently, two challenges arise: $(i)$ how to efficiently find the best locations to introduce adversarial perturbations; and $(ii)$ how to decide on the strength of the created perturbations. Although a few recent works (Zhu et al., 2020; Gan et al., 2020; Sankaranarayanan et al., 2017) look into this field, they either add perturbations in the input embeddings or all the intermediate features, yet have not reached a coherent conclusion.

To efficiently learn an optimal strategy of perturbation injection, we further propose a learnable adversarial feature augmentation (L-ALFA) framework, which is capable of automatically adjusting the position and strength of introduced feature perturbations. The proposed approach not only

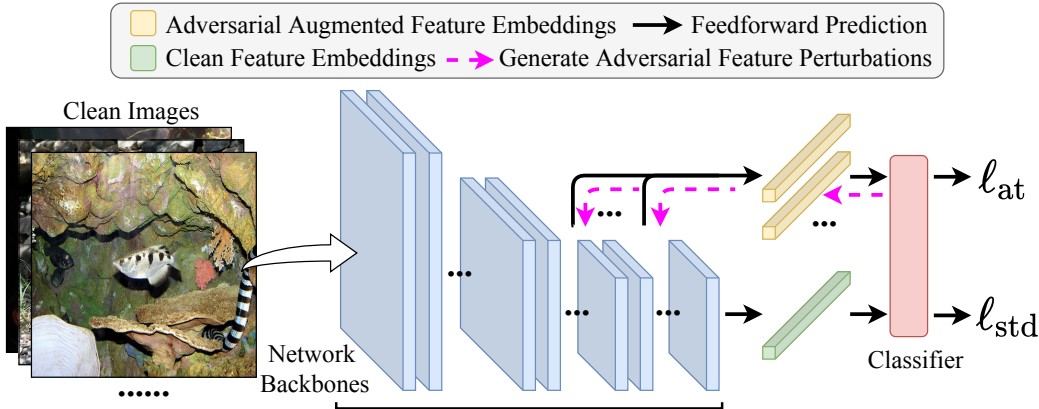

Figure 1: Overview of Adversarial Feature Augmentation for image recognition. From left to right, clean images are fed into network backbones to extract clean feature embeddings. Then, adversarial perturbations are generated to augment several intermediate features (in the direction of purple paths). In the end, both adversarial augmented and clean feature embeddings are taken as inputs by the classifer, and optimized by adversarial ($\mathcal{L}_{at}$) and standard training ($\mathcal{L}_{std}$) objectives.

circumvents laborious hyper-parameter tuning, but also fully unleashes the power of adversarial feature augmentation. Experiments show that this strategy gains a substantial performance margin over existing feature augmentation methods (Li et al., 2020). In addition, we find that learnable ALFA and exhaustively-tuned ALFA exhibit consistent patterns: applying *weak* adversarial feature augmentations to the *last* layers of deep neural networks can boost generalization performance.

The main contributions are summarized as follows. ($i$) We introduce a new approach of adversarial feature augmentation (ALFA) to improve the generalization ability of neural networks, which applies adversarial perturbations to the feature space rather than raw image pixels. ($ii$) To tackle the dilemma of laborious hyper-parameter tuning in generating adversarial features, we propose learnable adversarial feature augmentation (L-ALFA) to automatically tailor target perturbations and their locations. ($iii$) Comprehensive experiments on CIFAR-10, CIFAR-100, and ImageNet datasets across multiple backbone networks demonstrate the superiority of the proposed methods.

## 2 RELATED WORK

**Adversarial Training**   Deep neural networks are notoriously vulnerable to adversarial samples (Szegedy et al., 2013; Goodfellow et al., 2015b), which are crafted with malicious yet negligible perturbations (Goodfellow et al., 2015a; Kurakin et al., 2016; Madry et al., 2018a). In order to improve the robustness against adversarial samples, various defense mechanisms have been proposed (Zhang et al., 2019a; Schmidt et al., 2018; Sun et al., 2019; Nakkiran, 2019; Stutz et al., 2019; Raghunathan et al., 2019). Among these works, adversarial-training-based methods (Madry et al., 2018b; Zhang et al., 2019a) have achieved consistently superior performance in defending state-of-the-art adversarial attacks (Goodfellow et al., 2015a; Kurakin et al., 2016; Madry et al., 2018a). Although adversarial training substantially improves model robustness, it usually comes at the price of compromising the standard accuracy (Tsipras et al., 2019), which has been demonstrated both empirically and theoretically (Zhang et al., 2019a; Schmidt et al., 2018; Sun et al., 2019; Nakkiran, 2019; Stutz et al., 2019; Raghunathan et al., 2019).

Recently, researchers start to investigate improving clean set accuracy with adversarial training (Xie et al., 2020; Zhu et al., 2020; Wang et al., 2019a; Gan et al., 2020; Wei & Ma, 2019) (Ishii & Sato, 2019). Xie et al. (2020) shows that performance on the clean dataset can be enhanced by using adversarial samples with pixel-level perturbation generation. Zhu et al. (2020) and Wang et al. (2019a) apply adversarial training to natural language understanding and language modeling, both successfully achieving better standard accuracy. Gan et al. (2020) achieves similar success on many vision-and-language tasks. There also exist parallel studies that employ handcrafted or auto-

generated perturbed features to ameliorate generalization (Wei & Ma, 2019) (Ishii & Sato, 2019) or robustness (Sankaranarayanan et al., 2017).

However, two key issues remain unexplored: ($i$) which layers to introduce adversarial feature augmentations; ($ii$) how strong the perturbations should be. For the former, Zhu et al. (2020); Wang et al. (2019a); Gan et al. (2020) try to perturb the input embeddings of transformer models, while Wei & Ma (2019); Sankaranarayanan et al. (2017) insert perturbations to all layers of a convolutional network. Regarding the above issue, all the methods need arduous and heuristic tunings. In our paper, we present a different observation that augmenting the last layers' feature embeddings with weak adversarial feature perturbations can gain higher standard accuracy. The L-ALFA framework inspired by this observation can effectively alleviate laborious tuning, which otherwise is inevitable.

**Feature Augmentation**  Although pixel-level data augmentation techniques (Simard et al., 1993; Schölkopf et al., 1996) have been widely adopted, feature space augmentations have not received the same level of attention. A few pioneering works propose generative-based feature augmentation approaches for domain adaptation (Volpi et al., 2018), imbalance classification (Zhang et al., 2019b), and few-shot learning (Chen et al., 2019). Another loosely related field is feature normalization (Ioffe & Szegedy, 2015; Li et al., 2020). MoEx (Li et al., 2020) is a newly proposed method that can be regarded as a feature augmentation technique, which leverages the first and second-order moments extracted and re-injected by feature normalization. It is worth mentioning that all the approaches aforementioned are orthogonal to our proposed method, and can be combined for further generalization improvement, which is left as future work.

## 3    Adversarial Feature Augmentations (ALFA)

In the proposed ALFA framework, we generate adversarial perturbations in the intermediate feature embedding space, rather than applying perturbations to raw image pixels as in common practice. Thus, adversarial training can be formulated as an effective regularization to improve the generalization ability of deep neural networks.

### 3.1    Notations

Given a dataset $\mathcal{D} = \{\mathbf{x}, y\}$, where $\mathbf{x}$ is the input image and $y$ is the corresponding one-hot ground-truth label. Let $f(\mathbf{x}; \mathbf{\Theta})$ represents predictions of a deep neural networks, and $f_i(\mathbf{x}; \mathbf{\Theta}^{(i)})|_{i=1}^{r+1}$ is the intermediate feature embedding from the $i$-th layer. The $(r+1)$-th layer denotes the classifier, therefore $f_{r+1}(\mathbf{x}; \mathbf{\Theta}^{(r+1)}) = f(\mathbf{x}; \mathbf{\Theta})$. Adversarial training can be formulated as the following min-max optimization problem:

$$\min_{\mathbf{\Theta}} \quad \mathbb{E}_{(\mathbf{x},y)\in\mathcal{D}} \left[ \max_{||\boldsymbol{\delta}||_p \leq \epsilon} \mathcal{L}_{\mathrm{at}}(f(\mathbf{x}+\boldsymbol{\delta}; \mathbf{\Theta}); \mathbf{\Theta}; y) \right], \tag{1}$$

where $\boldsymbol{\delta}$ is the adversarial perturbation bounded by the $\ell_p$ norm ball, which is centered at $\mathbf{x}$ with radius $\epsilon$ which is the maximum perturbation magnitude. $\mathcal{L}_{\mathrm{at}}$ is the cross-entropy loss for adversarial training (AT). $\mathbb{E}_{(\mathbf{x},y)\in\mathcal{D}}$ takes the expectation over the empirical objective on the training dataset $\mathcal{D}$.

The inner optimization generates adversarial perturbation $\boldsymbol{\delta}$ via maximizing the empirical objective. It can be reliably solved by multi-step projected gradient descent (PGD) (Madry et al., 2018b) (without loss of generality, we take $||\cdot||_\infty$ perturbation as an example):

$$\boldsymbol{\delta}_{t+1} = \Pi_{||\boldsymbol{\delta}||_\infty \leq \epsilon} \left[ \boldsymbol{\delta}_t + \alpha \cdot \mathrm{sgn}(\nabla_{\mathbf{x}}\mathcal{L}_{\mathrm{at}}(f(\mathbf{x}+\boldsymbol{\delta}_t; \mathbf{\Theta}); \mathbf{\Theta}; y) \right], \tag{2}$$

where $t$ is the number of steps, $\alpha$ denotes the learning rate of inner maximization, sgn is the sign function, and $\mathcal{L}_{\mathrm{at}}$ is the adversarial training objective on adversarial images.

### 3.2    Perturbations in the Embedding Space via ALFA

Here, we extend the conventional adversarial perturbations to the feature embedding space. We start from the training objective of ALFA as follows:

$$\min_{\mathbf{\Theta}} \quad \mathbb{E}_{(\mathbf{x},y)\in\mathcal{D}} \left[ \mathcal{L}_{\mathrm{std}}(\mathbf{x}; \mathbf{\Theta}; y) + \lambda \cdot \sum_i \max_{||\boldsymbol{\delta}^{(i)}||_\infty \leq \epsilon} \mathcal{L}_{\mathrm{at}}(f_i(\mathbf{x}; \mathbf{\Theta}^{(i)}) + \boldsymbol{\delta}^{(i)}; \mathbf{\Theta}; y) \right], \tag{3}$$

where $\mathcal{L}_{\text{std}}$ is the cross-entropy (XE) loss on clean images, $\mathcal{L}_{\text{at}}$ here is the cross-entropy loss for adversarial training (AT) on adversarial augmented feature embeddings. $\lambda$ is the hyperparamter to control the influence of AT regularization, which is tuned by grid search. $\boldsymbol{\delta}^{(i)}$ is the adversarial perturbation on the feature of layer $i$, generated as follows:

$$\boldsymbol{\delta}_{t+1}^{(i)} = \Pi_{||\boldsymbol{\delta}||_\infty \leq \epsilon} \left[ \boldsymbol{\delta}_t^{(i)} + \alpha \cdot \text{sgn}(\nabla_{\mathbf{x}} \mathcal{L}_{\text{at}}(f_i(\mathbf{x}; \boldsymbol{\Theta}^{(i)}) + \boldsymbol{\delta}_t^{(i)}; \boldsymbol{\Theta}; y)) \right]. \tag{4}$$

It is worth noting that, for crafting $\boldsymbol{\delta}^{(i)}$, at each step, the gradient is only back-propagated to the $i$-th layer without going further, which is much more computationally efficient compared to generating perturbations in the input embedding space. In practice, we set the maximum magnitude of crafted feature perturbation $\epsilon$ to be unbounded, and the projected gradient descent will be replaced by gradient descent.

| Settings | Standard Accuracy (%) | |
| --- | --- | --- |
| | ResNet-20s | ResNet-56s |
| Baseline | 91.25 | 93.03 |
| $\alpha = \frac{0.5}{255}$ | **92.52** | 94.38 |
| $\alpha = \frac{1.0}{255}$ | 92.47 | 94.47 |
| $\alpha = \frac{1.5}{255}$ | 92.34 | **94.72** |
| $\alpha = \frac{2.0}{255}$ | 91.36 | 93.45 |

Table 1: Applying ALFA with different step size $\alpha$ to the feature embeddings from the last block of ResNet-20s/56s on CIFAR-10. Larger $\alpha$ indicates stronger feature perturbations.

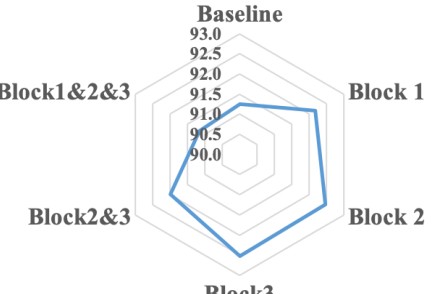

Figure 2: Applying ALFA to the feature embeddings from different blocks or the combinations of top-performing blocks. Experiments are conduted on CIFAR-10 with ResNet-20s.

In ALFA, the two most essential factors are: $(i)$ where to introduce adversarial perturbations; and $(ii)$ how strong the perturbations should be. Table 1 and Figure 2 present some preliminary results to understand this. Results shows that the performance of ALFA relies particularly on the location (*i.e.*, which blocks) and strength (*i.e.*, step size $\alpha$) of the introduced feature perturbations. An inadequate configuration (*e.g.,* applying ALFA to all blocks 1, 2 and 3 as shown in Figure 2) might cause accuracy degradation. More analyses are provided in Section 4.3. To determine the best configuration, we further design a learnable adversarial feature augmentation (L-ALFA) approach to automatically adjusting the location and strength of perturbations for best augmentation performance, which will be explained in the next sub-section.

## 3.3 LEARNABLE PERTURBATIONS VIA L-ALFA

To ascertain the two critical settings (locations and strength) of feature augmentations, we introduce an enhanced ALFA method, L-ALFA, which also eliminates laborious tuning. Specifically, in layer $i$, for the PGD-generated perturbation $\boldsymbol{\delta}^{(i)}$, we apply a learnable parameter $\eta_i$ to control the magnitude of $\boldsymbol{\delta}^{(i)}$ before adding it to the corresponding feature embeddings. Thus, a learned near-zero $\eta_i$ indicates that it is unnecessary to inject feature perturbations on layer $i$. Furthermore, we also introduce the $\ell_1$ sparsity constraint on the learnable perturbation magnitude $\boldsymbol{\eta}$. The design philosophy is that applying ALFA on all layers of deep neural network does not benefit (or even hurt) the standard accuracy, as exemplified in Figure 2 and Figure 3.

The optimization problem is then formulated as:

$$\min_{\boldsymbol{\Theta}, \{\eta_i\}_1^r, \boldsymbol{\eta} \in \mathcal{P}} \mathbb{E}_{(\mathbf{x}, y) \sim \mathcal{D}} \left[ \mathcal{L}_{\text{std}} + \lambda \cdot \sum_{i=1}^r \max_{||\boldsymbol{\delta}^{(i)}||_\infty \leq \epsilon} \mathcal{L}_{\text{at}}(f_i(\mathbf{x}; \boldsymbol{\Theta}^{(i)}) + \eta_i \cdot \boldsymbol{\delta}^{(i)}; \boldsymbol{\Theta}; y) + \gamma \cdot ||\boldsymbol{\eta}||_1 \right], \tag{5}$$

where $\mathcal{L}_{\text{std}} = \mathcal{L}_{\text{XE}}(\mathbf{x}; y; \boldsymbol{\Theta})$, $\mathcal{P} = \{\boldsymbol{\eta} | \mathbf{1}^{\text{T}} \boldsymbol{\eta} = 1\}$, $\boldsymbol{\eta} = (\eta_1, \eta_2, \cdots, \eta_r)$ is the learnable strength of feature perturbations, $\gamma$ is the hyperparameter to control the sparsity level. $\gamma$ can be chosen from $\{0.5, 1.0, 2.0\}$. To solve equation 5, we first generate feature perturbations $\boldsymbol{\delta}^{(i)}$ via multi-step PGD.

---

**Algorithm 1** Learnable Adversarial Feature Augmentation (L-ALFA).

1: Input: given $\boldsymbol{\Theta}_0, \boldsymbol{\eta}_0, \boldsymbol{\delta}_0$. (In our case, $\boldsymbol{\eta}_0 = (1, \cdots, 1) \in \mathbb{R}^r$)
2: **for** $n = 1, 2, \cdots, N$ iterations **do**
3:     Given $\boldsymbol{\Theta}_{n-1}, \boldsymbol{\eta}_{n-1}$, generate adversarial perturbation $\boldsymbol{\delta}_n = (\boldsymbol{\delta}_n^{(1)}, \cdots, \boldsymbol{\delta}_n^{(r)})$ via multi-step PGD;
4:     Given $\boldsymbol{\delta}_n$, perform SGD to update $\boldsymbol{\Theta}_n, \boldsymbol{\eta}_n$;
5:     Project $\boldsymbol{\eta}_n$ into $\mathcal{P}$ via the bisection method (Wang et al., 2019b).
6: **end for**

---

Then, we minimize the empirical training objective to update the network weights $\boldsymbol{\Theta}$ and $\boldsymbol{\eta}$ through stochastic gradient descent (SGD) (Ruder, 2016). In the end, we project $\boldsymbol{\eta}$ into $\mathcal{P}$ and repeat the above steps until the training converges. The full algorithm is summarized in Algorithm 1.

## 4 EXPERIMENTS

We conduct extensive experiments on multiple benchmarks to validate the generalization ability of ALFA and L-ALFA, evaluating across different backbone networks for image recognition. Ablation studies and analysis of the learned distribution of perturbation magnitude are also provided.

### 4.1 EXPERIMENTAL SETUP

**Datasets and Backbones** We consider three representative datasets: CIFAR-10, CIFAR-100 (Krizhevsky et al., 2009), and ImageNet (Deng et al., 2009). In our experiments, the original training datasets are randomly split into 90% training and 10% validation. The early stopping technique is applied to find the top-performing checkpoints on the validation set. Then, the selected checkpoints are evaluated on the test set to report the performance. From our observations, the hyperparameters are quite stable from validation to test sets. We evaluate large backbone networks (ResNet-18/50/101/152 (He et al., 2016)) on all three datasets, and test smaller backbones (ResNet-20s/56s) as well on CIFAR-10 and CIFAR-100. Ablation studies are implemented on CIFAR-10, where key observations can be generalized to other datasets.

**Training and Metrics** For network training on CIFAR-10 and CIFAR-100, we adopt an SGD optimizer with a momentum of $0.9$, weight decay of $5 \times 10^{-4}$, and batch size of $128$ for $200$ epochs. The learning rate starts from $0.1$ and decays to one-tenth at 50-th and 150-th epochs. We also perform a linear learning rate warm-up in the first $200$ iterations. For ImageNet experiments, following the official setting in Pytorch repository,[1] we train deep networks for $90$ epochs with a batch size of $512$, and the learning rate decay at 30-th and 60-th epoch. The SGD optimizer is adopted with a momentum of $0.9$ and a weight decay of $1 \times 10^{-4}$. We evaluate the generalization ability of a network with Standard Testing Accuracy (**SA**), which represents image recognition accuracy on the original clean test dataset.

### 4.2 EVALUATION AND ANALYSIS OF ALFA

For ALFA experiments, all hyperparameters are tuned by grid search, including PGD steps, step size $\alpha$, and the layers to introduce adversarial perturbations. For generated adversarial feature embeddings, we set the maximum perturbation magnitude $\epsilon$ to be unbounded [2], since there are no explicit constraints for feature perturbations, and the effect of tuning $\epsilon$ can be absorbed by adjusting PGD steps and step size $\alpha$.

Table 2 presents the standard testing accuracy of different models on CIFAR-10. Comparing the standard training with our proposed ALFA, here are the main observations:

($i$) ALFA obtains a consistent and substantial improvement over standard accuracy, *e.g.*, $1.27\%$ on ResNet-20s, $1.69\%$ on ResNet-56s, and $0.51\%$ on ResNet-50. This suggests that training with augmented features generated by ALFA effectively enhances the generalization of

---

[1]https://github.com/pytorch/examples/tree/master/imagenet
[2]In practice, the magnitude of crafted feature perturbation steadily stays in a range from $0.97$ to $1.10$ under the $\ell_2$ norm. Adversarial perturbations usually are applied to the normalized feature from batch normalization.

Table 2: Standard testing accuracy (%) on CIFAR-10 dataset. Standard Training stands for the traditional training with only clean data. We utilize PGD-5 to generate the adversarial perturbations on the feature embeddings from the last residual block of ResNets. $\epsilon$ is unbounded here. Step size $\alpha = 0.5/255$ for ResNet-20s, $\alpha = 1.0/255$ for ResNet-18 and ResNet-101, and $\alpha = 1.5/255$ for ResNet-56s and ResNet-50. ↑ indicates the improvement over SA compared to the corresponding baseline in standard training.

| Settings | ResNet-20s | ResNet-56s | ResNet-18 | ResNet-50 | ResNet-101 |
|---|---|---|---|---|---|
| Standard Training | 91.25 | 93.03 | 94.30 | 94.73 | 95.17 |
| ALFA | 92.52 (↑ 1.27) | 94.72 (↑ 1.69) | 94.65 (↑ 0.35) | 95.24 (↑ 0.51) | 95.38 (↑ 0.21) |

Table 3: Standard testing accuracy (%) on CIFAR-100 and ImageNet datasets. For CIFAR-100, we apply PGD-5 to augment the feature embeddings in the last block with adversarial perturbations. $\epsilon$ is unbounded, and $\alpha = 0.5/255$ for both ResNet-20s and ResNet-56s. As for ImageNet dataset, one-step PGD is selected for ALFA, with $\epsilon$ unbounded and $\alpha = 0.5/255$. All perturbations are applied to feature embeddings in the last block.

| Settings | CIFAR-100 | | ImageNet | | | |
|---|---|---|---|---|---|---|
| | ResNet-20s | ResNet-56s | ResNet-18 | ResNet-50 | ResNet-101 | ResNet-152 |
| Standard Training | 66.92 | 71.22 | 69.38 | 75.21 | 77.10 | 78.31 |
| ALFA | 67.79 (↑ 0.87) | 72.36 (↑ 1.14) | 70.19 (↑ 0.81) | 76.23 (↑ 1.02) | 78.04 (↑ 0.94) | 78.65 (↑ 0.34) |

deep neural networks. We hypothesize that this is because adversarial perturbed features are treated as an implicit regularization, leading to better solutions for network training.

(ii) Different backbones prefer diverse strengths of adversarial feature perturbations, and there is no obvious pattern. For example, networks in middle size, such as ResNet-56s and ResNet-50, tend to favor larger perturbations (i.e., $\alpha = 1.5/255$) compared to small ResNet-20s ($\alpha = 0.5/255$) and deep ResNet-101 ($\alpha = 1.0/255$).

(iii) Shallow ResNets benefit more from ALFA than deep ResNets (e.g., $1.27\%$ on ResNet-20s vs. $0.21\%$ on ResNet-101). A possible reason is that the performance of standard trained ResNets is already saturated on the small-scale CIFAR-10 dataset, leaving little room for improvement.

Results on CIFAR-100 and ImageNet are summarized in Table 3. We observe that ALFA consistently boosts the generalization ability of multiple ResNets on both CIFAR-100 and ImageNet, e.g., $1.14\%$ for ResNet-56s on CIFAR-100, $1.02\%$ for ResNet-50 on ImageNet. Furthermore, we notice that ALFA advocates different steps of PGD to achieve superior performance on diverse datasets. To fully understand these sensitive yet critical factors, we conduct a systematical and comprehensive ablation study in the next sub-section.

### 4.3 ABLATION STUDIES

**Strength and Locations of ALFA** To understand the effect or the strength of injected adversarial perturbations, we implement ResNet-18 on CIFAR-10 and examine the performance across different step sizes and numbers of PGD steps. Table 4 shows that perturbing with step size $\alpha = 1.0/255$ obtains a larger gain by $0.35\%$ SA. In addition, excessive weak (e.g., $\alpha = 0.5/255$) or strong (e.g., $\alpha = 1.5/255$) adversarial feature augmentations may incur performance degradation. For the ablation of PGD steps, we implement ResNet-18 on ImageNet as well. Table 5 demonstrates that ALFA with PGD-5 and PGD-1 works the best for CIFAR-10 and ImageNet, respectively, indicating that the strength of generated perturbations is an essential and sensitive hyperparameter for ALFA.

Then, we analyze the effect of locations (i.e., where to apply ALFA) via two typical backbones: ResNet-56s on CIFAR-10 and ResNet-18 on ImageNet. In each setting, we present a detailed analysis on which layer and how many layers the feature embeddings should be adversarially augmented for achieving the best performance. Figure 3 presents the layer preference of feature perturbations when applying ALFA to different blocks or some combinations of blocks. We notice that introducing ALFA to the last block achieves better standard accuracy, while the performance deteriorates

Table 4: Standard testing accuracy (%) on CIFAR-10 dataset. We perturb the feature embeddings in the last block of ResNet-18 via PGD-5 and diverse step size $\alpha$. For reference, SA of standard trained model is 94.30%. ↑/↓ indicates SA improvement/degradation compared to the corresponding baseline in standard training.

| Step size $\alpha$ | $\frac{0.5}{255}$ | $\frac{1.0}{255}$ | $\frac{1.5}{255}$ | $\frac{2.0}{255}$ | $\frac{4.0}{255}$ |
|---|---|---|---|---|---|
| ALFA | 93.15 (↓ 1.15) | 94.65 (↑ 0.35) | 94.34 (↑ 0.04) | 94.36 (↑ 0.06) | 93.30 (↓ 1.00) |

Table 5: Standard testing accuracy (%) on CIFAR-10 and ImageNet datasets. For CIFAR-10, we perturb the feature embeddings in the last block of ResNet-18, via PGD-1/3/5/7/10 with step size $\alpha = 1.0/255$. As a reference, the SA of standard trained model is 94.30%. As for ImageNet, we also perturb the last block features of ResNet-18 via PGD-1/3/5 and step size $\alpha = 0.5/255$. The reference SA is 69.38%.

| Steps | CIFAR-10 | | | | | ImageNet | | |
|---|---|---|---|---|---|---|---|---|
| | PGD-1 | PGD-3 | PGD-5 | PGD-7 | PGD-10 | PGD-1 | PGD-3 | PGD-5 |
| ALFA | 94.46 (↑ 0.16) | 94.43 (↑ 0.13) | 94.65 (↑ 0.35) | 94.39 (↑ 0.09) | 94.17 (↓ 0.13) | 70.19 (↑ 0.81) | 68.65 (↓ 0.73) | 67.42 (↓ 1.96) |

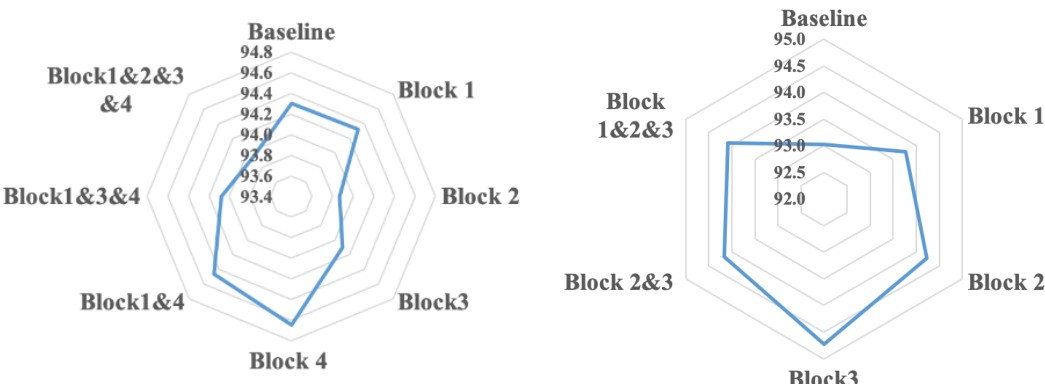

Figure 3: Standard testing accuracy (%) of ResNet-18 (Left) and ResNet-56s (Right) on CIFAR-10 dataset, with ALFA applied to different blocks and several combinations of top-performing blocks. 'Block' represents the residual blocks in the ResNets. Block 4 of ResNet-18 and Block 3 of ResNet-56s are the last blocks before the classifiers. The adversarial perturbations on each block's feature embeddings are generated by carefully tuned configurations, including PGD-5, unbounded $\epsilon$ and step size $\alpha \in \{\frac{0.5}{255}, \frac{1.0}{255}, \frac{1.5}{255}\}$.

after injecting ALFA to multiple blocks. These results demonstrate that the strength and location for ALFA play a crucial role and need to be cautiously selected, which motivates us to design the learnable framework, L-ALFA.

**ALFA vs. Other Feature Augmentations** One natural baseline is adding random noise to the feature embeddings. For each training iteration, a new random noise sampled from a Gaussian distribution $\mathcal{N}(0, (1.0/255)^2)$ is applied to the same feature embeddings. Another representative feature augmentation recently proposed is MoEx[3] (Li et al., 2020), which is compared as another baseline. For ALFA, we choose the best hyperparameter configurations for ResNet-56s and ResNet-18 on CIFAR-10, i.e., perturbing the last block feature embeddings with PGD-5, and step size ($\alpha = \frac{1.5}{255}$ for ResNet-56s and $\alpha = \frac{1.0}{255}$ for ResNet-18). As shown in Table 7, ALFA significantly surpasses MoEx and random-noise-based feature augmentation, demonstrating that feature perturbations generated from ALFA are non-trivial.

**ALFA vs. AdvProp** We compare ALFA with AdvProp (Xie et al., 2020) on CIFAR-10 with ResNet-18 and on ImageNet with EfficientNet-B0 (Tan & Le, 2019), as presented in Table 6. Training on a single GTX1080 Ti GPU for CIFAR-10 experiments, AdvProp achieves 94.52% accuracy with 123 seconds per epoch; ALFA obtains 94.65% with 28 seconds per epoch, where standard training takes 23 seconds per epoch. The experiments on ImageNet (batch size 256) are conducted on 2 Quadro RTX 6000 GPUs with 24G×2 memory in total, and the reported running time is per epoch.

---

[3]We implement MoEx based on their released repository, https://github.com/Boyiliee/MoEx

Table 6: Running time per epoch and standard accuracy (SA) comparison across standard training, AdvProp, and ALFA.

| Settings | ResNet-18 on CIFAR-10 | | | EfficientNet-B0 on ImageNet | | |
|---|---|---|---|---|---|---|
| | Standard Training | AdvProp | ALFA | Standard Training | AdvProp | ALFA |
| SA (%) | 94.30 | 94.52 (↑ 0.22) | 94.65 (↑ 0.35) | 77.00 | 77.50 (↑ 0.50) | 77.60 (↑ 0.60) |
| Time | 23s | 28s | 123s | 2628s | 3140s | 13352s |

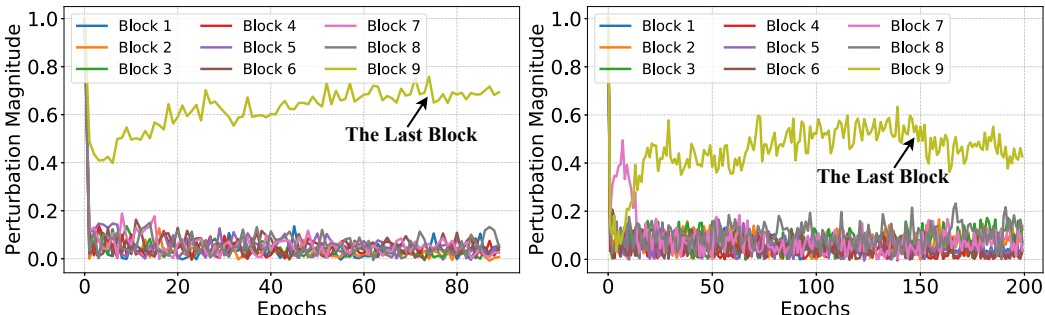

Figure 4: The learned distribution of perturbation magnitude vector $\eta$ in L-ALFA over training epochs on ImageNet (Left) and CIFAR-10 (Right). The curves of the last block (*i.e.*, Block 9) feature embeddings are highlighted by black arrows.

Table 8: Standard testing accuracy (%) of L-ALFA with different ResNets ($\dim(\eta) = 9$). Performance differences are computed between L-ALFA/ALFA and Standard Training.

| Datasets | CIFAR-10 (ResNet-18) | CIFAR-100 (ResNet-56s) | ImageNet (ResNet-50) |
|---|---|---|---|
| Standard Training | 94.30 | 71.22 | 75.21 |
| L-ALFA | 94.62 (↑ 0.32) | 72.40 (↑ 1.18) | 76.12 (↑ 0.91) |
| ALFA | 94.65 (↑ 0.35) | 72.36 (↑ 1.14) | 76.23 (↑ 1.02) |

As shown in Table 6, ALFA obtains a similar performance improvement with less computational cost, compared with pixel-level adversarial augmentations (e.g., AdvProp).

**Robust Performance of ALFA** Although the robust testing accuracy (RA) is not the focus of ALFA, we report it for completeness. We implement the standard, ALFA and the adversarial trained ResNet-18 networks on CIFAR-10. The adversarial trained model uses PGD-10 with step size $\alpha = \frac{2}{255}$ and $\epsilon = \frac{8}{255}$ for training. Then, PGD-20 with the same $\alpha$ and $\epsilon$ is applied to evaluate the robust performance of the three models. We observe that ALFA trained models (4.86% RA) yield moderate robustness, compared to models from standard (0.00% RA) and adversarial (50.72% RA) trained models.

Table 7: Standard testing accuracy of ALFA vs. other methods. All experiments are repeated for 5 runs, with errorbars of one standard deviation reported.

| Methods | ResNet-56s | ResNet-18 |
|---|---|---|
| Random Noise | 93.17 ± 0.09 | 94.34 ± 0.10 |
| MoEx | 92.90 ± 0.33 | 94.16 ± 0.23 |
| ALFA | **94.72 ± 0.06** | **94.65 ± 0.08** |
| AdvProp | - | 94.52 ± 0.28 |

### 4.4 EVALUATION AND ANALYSIS OF L-ALFA

Results on L-ALFA are presented in Table 8 and Figure 4. L-ALFA consistently improves the generalization of trained networks by 0.32% on CIFAR-10, 1.18% on CIFAR-100, and 0.91% on ImageNet. Although the achieved performance is close to ALFA, L-ALFA saves toilsome tuning by automatically adjusting the strength and locations of adversarial feature augmentations. Another interesting finding is that L-ALFA automatically learns the effective trick of only perturbing the last block feature embeddings, which is consistent with our observations made from ALFA.

**Number of Perturbed Layers**
To study the effect of the dimension of the learnable perturbation magnitude $\boldsymbol{\eta}$ in L-ALFA, we implement ResNet-18 on CIFAR-10 for the additional experiment. ResNet-18

Table 9: Ablation studies on the dimension of $\boldsymbol{\eta}$. SA (%) on CIFAR-10 dataset is reported.

| Dimension of $\boldsymbol{\eta}$ | $\dim(\boldsymbol{\eta}) = 4$ | $\dim(\boldsymbol{\eta}) = 9$ | $\dim(\boldsymbol{\eta}) = 20$ |
|---|---|---|---|
| L-ALFA | 94.52 (↑ 0.22) | 94.62 (↑ 0.32) | 93.54 (↓ 0.76) |

has four residual blocks and twenty convolution layers, and perturbing them will result in $\dim(\boldsymbol{\eta})$ being equal to 4 and 20, respectively. We also try introducing adversarial feature augmentations to only some of the intermediate layers, such as perturbing the features after each skip connection (*i.e.*, $\dim(\boldsymbol{\eta}) = 9$). As shown in Table 9, an unduly high dimension of $\boldsymbol{\eta}$, *i.e.*, perturbing features in almost all the layers, is harmful to model generalization. Therefore, learning feature perturbations by blocks or by skip connections is adequate for L-ALFA.

Table 10: Ablation studies of the $\ell_1$ regularization. SA (%) on CIFAR-100 dataset is reported.

| $\gamma$ | 0.1 | 0.2 | 0.5 | 1.0 | 2.0 | 10.0 |
|---|---|---|---|---|---|---|
| SA of L-ALFA | 70.68 (↑ 0.54) | 72.13 (↑ 0.91) | 72.35 (↑ 1.13) | 72.40 (↑ 1.18) | 72.38 (↑ 1.16) | 71.92 (↑ 0.70) |

**Ablation of the $\ell_1$ regularization in L-ALFA** Results of ResNet-56s on CIFAR-100 are collected in Table 10. We observe that $\gamma$ can be roughly chosen from $\{0.5, 1.0, 2.0\}$ and obtain similar competitive performance. Excessive large or small values of $\gamma$ obtain less performance improvements or even incur degradation.

## 5 CONCLUSION

In this paper, we present ALFA, an advanced adversarial training framework for image recognition. By applying adversarial perturbations on feature embeddings, and by jointly training with both clean and adversarial augmented features, ALFA improves the generalization of diverse neural network backbones across multiple image recognition datasets such as ImageNet. Systematical ablation studies reveal that introducing weak adversarial feature augmentations to the last layers of networks contributes more, which is different from previous findings. Furthermore, we propose L-ALFA to learn a better augmentation, and also save the laborious tuning on ALFA. For future work, we plan to extend the ALFA to other vision tasks, such as object detection and semantic segmentation.

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
