# OpenReview forum: "ALFA: Adversarial Feature Augmentation for Enhanced Image Recognition"
_ICLR.cc/2021/Conference — Reject_

### Official Review · AnonReviewer3 · 2020-10-25

**Rating:** 5
**Confidence:** 3

**Review:**

Overview of paper: this work tackles the task of adversarial augmentation for better generalization. Instead of augmentation the pixels space, which is expensive and potentially harder, they augment the intermediate feature representation. As the choice of the particular layer for application of the perturbations affects performance, the authors, optimize it jointly with the rest of the parameters. Experiments show this method  improves accuracy over standard training.

Novelty: although adversarial training on raw image is known to improve generalization, doing so on image features is novel as far as I am aware. Additionally, jointly choosing the layers to be perturbed is also new in my understanding (although the main benefit is in the analysis, as the fixed strategy of perturbing the last block seems comparable).

Evaluation: The proposed feature adversarial training seems to consistently improve generalization on several popular datasets, however there are a few limitations: i) the gap is not huge. ii) it is not clear that the difference is significant from Xie et al. iii) nearly all results are on Cifar10, including the baseline comparisons iv) the speed up due to operating on features rather than pixels is cited as the main motivation but limited effort exists to evaluate it.

Presentation: the paper is nicely written, and is easy to follow.

Other questions: did you use auxiliary BNs like Xie et al? In what experiments did ALFA-L beat ALFA on the last block? (Sec 4.4 is a bit hazy there)

Overall: Adversarial feature perturbation for generalization is an interesting ideas and was shown to have some benefits. The evaluation of accuracy and runtime against other reasonable methods (particularly doing the same on pixels) is limited. I am positively inclined towards this paper and hope the authors can address by concerns in the rebuttal.

##########################################################################################

The response addressed some of my concerns, but I am concerned about L-ALFA taking such a large part of the paper and then shown to not help so much over just picking the final laye nor being much faster than the baseline. The more important ALFA hyperparameters that would most benefit from automatic tuning are not sufficiently treated. Although I do like the objective of this paper and some of the approaches, I think it might need to be revised and resubmitted, incorporating the extensive discussion presented by all the reviewers.

---

> ### Author Response · Authors · 2020-11-19
> **Response to Reviewer #3 [Weakness 1-3]**
>
> Thanks for your insightful comments. Below, we provide detailed responses to your concerns.
>
> [Weakness 1: The gap is not huge.]
>
> It is fair to note that these datasets are highly competitive, and it is impressive to obtain further improvements based on baselines’ almost saturated performance. Our proposed methods ALFA, as an implicit data augmentation or regularization, achieves consistent generalization improvements over strong baselines on CIFAR-10, CIFAR-100, and ImageNet benchmarks across different backbone networks for image recognition.
>
> [Weakness 2: The difference from Xie et al..]
>
> Our methods (ALFA, L-ALFA) have several key differences from AdvProp in Xie et al.
>
> (1) Our methods leverage adversarial perturbations in the feature space to improve generalization, instead of generating computational expensive pixel-level perturbations on raw images (Xie et al).
>
> (2) To efficiently learn an optimal strategy of perturbation injection, we further propose a learnable adversarial feature augmentation (L-ALFA) framework, which is capable of automatically adjusting the position and strength of introduced feature perturbations. This is not covered in Xie et al.
>
> (3) We do not use the auxiliary batch normalization, which is the main contribution in Xie et al.
>
> [Weakness 3: Nearly all results are CIFAR-10.]
>
> We respectfully do not agree. We verify our proposed ALFA on CIFAR-100 with ResNet-20s and ResNet-56s backbones (Table 3), on ImageNet with ResNet-18, ResNet-50, ResNet-101, and ResNet-152 backbones (Table 3). We also report the ablation of PGD steps on ImageNet (Table 4). In addition, we validate L-ALFA on both CIFAR-100 and ImageNet (Table 7). Note that, in both Table 3 and Table 4, we include the comparison with baselines.
>
> To further address your concern, we also provide the EfficientNet-B0 results on ImageNet here, Baseline (77.04%) vs. ALFA (77.48%). Extensive experiments demonstrate that ALFA achieves consistent generalization improvement over multiple backbones and diverse datasets.

---

> > ### Comment · AnonReviewer3 · 2020-11-21
> > **Response**
> >
> > I thank the authors for taking the time to write such detailed responses and running the extra experiments! I have also read the other reviews and have some remaining concerns.
> >
> > **Utility of L-ALFA:** I am a little confused about the claim that L-ALFA improves results consistently. If I understand, it actually reduced results over just ALFA on the final layer on Cifar10 (ALFA +0.35 L-ALFA +0.32) and ImageNet (ALFA +1.02 L-ALFA +0.91), it did help on Cifar100 but by a small amount . Choosing just the final layer is a conscious design choice but a consistent one - I do not really understand why this is such a major concern that justifies the more complex and much slower L-ALFA. It does make ALFA a very simple approach - but I think that's a good thing. If my current understanding of the paper is correct, the current presentation of the paper is confusing - , there should be clearer recommendations to simply use ALFA.
> >
> > **Hyper-parameters and speed comparisons:** As mentioned by the other reviewers, the method is quite sensitive to hyperparameters. The specific training time of the method should be multiplied by the number of hyperparameters it would need to be evaluated to obtain top performance. This might compare less favourably with the baseline. Also, the numbers show that the runtime improvement of L-ALFA over AdvProp is marginal.
> >
> > **Number and size of steps vs. number of layers:** This paper spent a lot of effort trying to overcome the technical challenge of choosing the best layers, but it seems that the best choice is nearly always the last layer. On the other hand the adversarial training parameters seem to vary more between datasets even if just focusing on the last layer, but L-ALFA is less focused on removing this cost. Can the authors explain this choice of focus? Can they perceive a way to more efficiently learn the size and number of steps, just on the last layer?
> >
> > **Results on Cifar10:** I thank the authors for clarifying this point, but the numbers for top baseline AdvProp were only provided for Cifar10 - (thank you for providing it for EfficientNet-B0!). It would have been very helpful to see it for all the other models, given that the relative performance between ALFA and AdvProp seems to be mixed.

---

> > > ### Author Response · Authors · 2020-11-23
> > > **Response to Reviewer #3 [Feedback 1-3]**
> > >
> > > Thanks for your constructive feedback. We truly value your acknowledgment of our efforts in the extra experiments. Below, we provide detailed responses to your concerns.
> > >
> > > [Feedback1: Utility of L-ALFA.]
> > >
> > > Sorry for the confusion, and we will further clarify it in the updated draft. The claim in Section 4.4 is that L-ALFA consistently improves the generalization of vanilla trained networks, where the comparisons are between L-ALFA and Baselines rather than ALFA. Although ALFA is a simpler and effective approach, L-ALFA remains its necessities from the following three perspectives:
> > >
> > > - Although there exists a consistent observation that applying ALFA to the last layer is the top-performing setting across multiple datasets and diverse backbone networks, it does not mean this conclusion will hold for any unseen dataset and networks. As shown in Figure 2, the achieved performance of applying ALFA to Block2 (i.e., an intermediate layer) and Block3 (i.e., the last layer)  are almost the same. In contrast, L-ALFA is capable of automatically learning a satisfying configuration for adversarial feature augmentations under new scenarios.
> > >
> > > - It is worth mentioning that even if we adopt the “last layer” prior, it still needs arduous tuning for an appropriate attack strength for ALFA. Our proposed L-ALFA eliminates laborious tuning by automatically adjusting the perturbation magnitude of each perturbed feature. L-ALFA is much more user-friendly for beginners who have little hyperparameter tuning experience in the machine learning community.
> > >
> > > - As presented in Tables 3 and 7, L-ALFA moderately surpasses ALFA on CIFAR-100 with ResNet-56s, which demonstrates there exist better-performing layer combinations (e.g., 0.14\*Block7+0.15\*Block8 + 0.42\*Block9) for introducing feature perturbations compared to the single last layer option.
> > >
> > > [Feedback2: Utility of Hyper-parameters and Speed Comparisons.]
> > >
> > > We respectfully do not agree. Actually, compared with AdvProp [1], ALFA only has one extra hyperparameter (i.e., the location to apply ALFA). If we adopt the “last layer” prior, ALFA almost has the same hyperparameter tuning complexity as AdvProp’s. Moreover, results in Tables 1, 4, 5, and Figures 2, 3, show a large safe zone of hyperparameters that enable the effectiveness of ALFA.
> > >
> > > We agree that if there are dozens of layers that need to be perturbed, the running time improvements of L-ALFA may be marginal compared to AdvProp. Fortunately, there is the “last layer” prior, as mentioned by reviewer #3. Therefore, a possible more efficient L-ALFA implementation is only considering the last few layers since they are the top-performing candidates.
> > >
> > > [Feedback3: Number and Size of Steps v.s. Number of Layers.]
> > >
> > > Thanks for the insightful questions. We provide justifications as follows:
> > >
> > > - The choice of focus. On the one hand, although there exists a consistent observation that applying ALFA to the last layer is the top-performing setting across multiple datasets and diverse backbone networks, it does not mean this conclusion will hold for any unseen dataset and networks. As presented in Tables 3 and 7, L-ALFA moderately surpasses ALFA on CIFAR-100 with ResNet-56s, which demonstrates L-ALFA is capable of learning better-performing layer combinations than the single last layer option. On the other hand, L-ALFA can also automatically learn the attack strength by adjusting the perturbation coefficient $\eta$. Specifically, for PGD-1, the effect of tuning the step size $\alpha$ can be absorbed by adjusting $\eta$.
> > >
> > > - An efficient way to learn the size and number of steps just on the last layer. This is an interesting and constructive question! For PGD-1, the learning of tuning the step size $\alpha$ can be equivalently replaced by learning $\eta$. As for multi-step PGD processes, the step size $\alpha$ and PGD steps can no longer be formulated as a differentiable optimization as depicted in Equation 5. In this sense, a potential solution is to learn a reinforcement learning agent [2] for predicting the step size $\alpha$ and PGD steps. We will continue to explore this interesting and promising direction in the future.
> > >
> > > [1] Adversarial examples improve image recognition, CVPR 2020.
> > >
> > > [2] Automated Synthetic-to-Real Generalization, ICML 2020.

---

> > > ### Author Response · Authors · 2020-11-23
> > > **(Continued) Response to Reviewer #4 [Feedback 4]**
> > >
> > > [Feedback4: Results on CIFAR-10.]
> > >
> > > We sincerely appreciate that you value our efforts on the new experiments. Due to the limited time and computation resources, it is unaffordable to compare with AdvProp on more huge models (e.g., EfficientNet-B7). To further address your concerns, we conduct more experiments on CIFAR-100 and ImageNet with ResNet-56s and ResNet-50 respectively. Comprehensive comparison results are collected in below Table S3.
> > >
> > > Table S3 Comparison results across Baseline (Vanilla Training), ALFA, and AdvProp [1]. We conduct experiments on CIFAR-10, CIFAR-100, and ImageNet with ResNet-18, ResNet-56s, ResNet-50, and EfficientNet-B0 respectively.
> > >
> > > |Datasets|CIFAR-10|CIFAR-100|ImageNet|ImageNet|
> > > |:-:|:-:|:-:|:-:|:-:|
> > > |Architectures|ResNet-18|ResNet-56s|ResNet-50|EfficientNet-B0|
> > > |Baseline|94.30%|71.22%|75.21%|77.0%|
> > > |ALFA|94.65% (↑0.35%)|72.36%(↑1.14%)|76.23%(↑1.02%)|77.5%(↑0.5%)|
> > > |AdvProp|94.52% (↑0.22%)|71.63 (↑0.41%)|75.89(↑0.68%)|77.6%(↑0.6%)|
> > >
> > > Please let us know if you have any additional concerns. And we are happy to further address them. Thank you!
> > >
> > > [1] Adversarial examples improve image recognition, CVPR 2020.
> > >
> > > [2] Automated Synthetic-to-Real Generalization, ICML 2020.

---

> ### Author Response · Authors · 2020-11-19
> **(Continued) Response to Reviewer #3 [Weakness 4 & Other Questions 1-2]**
>
> [Weakness 4: Need Extra Runtime Comparisons.]
>
> Thanks for your suggestions. To verify the claim, we provide runtime analyses as follows.
>
> Intuitively, the feature perturbation generation only requires backpropagation through a section of the network, while generating pixel-level perturbations needs backpropagation of the whole network. Meanwhile, as shown in Table 5, for ImageNet, single-step PGD is enough to craft effective adversarial feature augmentation with ALFA, while the top-performing pixel-level augmentation usually requires multi-step PGD.
>
> Empirically, for ALFA, we only apply adversarial perturbation to a single layer (usually the last layer or block), which allows ALFA to maintain cost efficiency even for deep networks. The efficiency of L-ALFA, compared to pixel-based methods, mainly depends on the number of perturbed layers (i.e., the dimension of $\eta$ in equation 5), and it is not necessary to perturb every layer. If there are hundreds of layers that need to be perturbed, L-ALFA may not be efficient enough. Fortunately, as shown in Table 8, a moderate number of perturbed layers (e.g., dim($\eta$)=9) is enough to exhibit the effectiveness of L-ALFA. We provide an updated time comparison with ResNet-18 per training epoch here: AdvProp (123s) vs. ALFA (28s) vs. L-ALFA with dim($\eta$)=9 (91s) vs. Standard training (23s).
>
> In addition, we also provide the comparison with EfficientNet-B0 on ImageNet, across i) standard training baseline, ii) feature-level adversarial augmentation, ALFA (applied to the last layer), iii) pixel-level adversarial augmentation, AdvProp in the following Table S3. The experiments (batch size 256)  are conducted on 2 Quadro RTX 6000 GPUs with 24G*2 memory in total, and the reported running time is per epoch. As shown in Table S3, ALFA obtains a similar performance improvement with less computational cost, compared with pixel-level adversarial augmentations (e.g., AdvProp).
>
>
> |Table S3|Baseline|ALFA|AdvProp|
> |:-:|:-:|:-:|:-:|
> |Accuracy|77.0%|77.5%(↑0.5%)|77.6%(↑0.6%)|
> |Running Time|2628s|3140s|13352s|
>
>
> [Other Question 1-2: Do we use auxiliary BN? L-ALFA versus ALFA.]
>
> We do not use the auxiliary BN.
>
> Sorry for the confusion. We will add a row in Table 7, representing the results from ALFA and making a direct comparison between ALFA and L-ALFA. We observe from Table 2, 3, and 7,   (ALFA v.s. L-ALFA) consistently improves the generalization of trained networks by (0.32% v.s. 0.35%) on CIFAR-10, (1.18% v.s. 1.14%) on CIFAR-100, and (0.91% v.s. 1.02%) on ImageNet. Note that here ALFA is carefully tuned yet L-ALFA are automatically learned. The main goal of L-ALFA is to eliminate laborious tuning of key parameters such as locations and strength of feature augmentations, and meanwhile achieves comparable performance improvements to ALFA.

---

### Official Review · AnonReviewer1 · 2020-10-27
**Although the motivation of this study is clear, the proposed method is not appropriately designed along with the motivation.**

**Rating:** 4
**Confidence:** 4

**Review:**

--Paper summary--

The authors propose Adversarial Feature Augmentation (ALFA), which augment features at hidden layers by adding adversarial perturbations. Where and how strongly the augmentation is conducted is automatically optimized via training. Experimental results show that the proposed method consistently improves the performance of baselines over several datasets and network architectures.

--Review summary--

Although the motivation of this study is clear, the proposed method is not appropriately designed along with the motivation. Moreover, its novely is merginal. I vote for rejection.

--Details--

Strength

- The motivation is clear and seems reasonable. Training with adversarial perturbations is known to be effective but computationally expensive. It can be problematic when the model or training data is large-scale.
- The proposed method consistently improves the performance of baselines over several datasets and network architectures.

Weakness and concerns

- Is the computational complexity of the proposed method really small? Since the adversarial perturbation is computed for every layer, its computational complexity should be almost same with that of standard adversarial training.
- The training objective shown in Eq. (3) is not reasonable. Since the norm of \delta is upper-bounded by a certain constant \epsilon, the effect of the adversarial perturbation can be reduced just by increasing the scale of features. Are features normalized ones?
- The optimization of \eta in L-ALFA is not reasonable. Since min_\eta comes after max_\delta, L-ALFA should choose the layer that corresponds to the smallest increase of loss by adding adversarial perturbation. Therefore, this design minimizes the effect of the augmentation, which is contradictive to the motivation of introducing \eta. Moreover, since \epsilon is common for all layers, the optimal \eta should be sensitive to the scale of features, which indicates that the performance of the proposed method would heavily depend on both how to initialize the model and whether any normalization is conducted in the model or not.
- Marginal novelty. An idea of adversarially augmented features has been already presented in [R1].
[R1] "Training Deep Neural Networks with Adversarially Augmented Features for Small-scale Training Datasets," IJCNN 2019.

---

> ### Author Response · Authors · 2020-11-14
> **Response to Reviewer #1 [Weakness 1-4]**
>
> Thanks for your insightful comments. Below, we provide detailed responses to your concerns.
>
> [Weakness 1: Computational complexity.]
>
> For ALFA, we only apply adversarial perturbation to a single layer, which allows ALFA to maintain cost efficiency even for deep networks. Note that the feature perturbation generation only requires backpropagation through a section of the network. The efficiency of L-ALFA, compared to pixel-based methods, mainly depends on the number of perturbed layers (i.e., the dimension of $\eta$ in equation 5), and it is not necessary to perturb every layer. We agree that if there are hundreds of layers that need to be perturbed, L-ALFA may not be efficient enough. Fortunately, as shown in Table 8, a moderate number of perturbed layers (e.g., dim($\eta$)=9) is enough to exhibit the effectiveness of L-ALFA. We provide an updated time comparison with ResNet-18 per training epoch here: AdvProp (123s) vs. ALFA (28s) vs. L-ALFA with dim($\eta$)=9 (91s) vs. Standard training (23s).
>
> [Weakness 2: Epsilon upper-bound.]
>
> As mentioned in Section 4, we adopt unbounded perturbation generation without the epsilon constraint. We will update our manuscript to clarify the confusion. Thanks for pointing out the missing details. Yes, we apply feature perturbations to the normalized feature from batch normalization. To further address your concern, here we report the magnitude of crafted feature perturbation in practice, which steadily stays in the range of 0.97 to 1.10 under the $\ell_2$ norm.
>
> [Weakness 3: The optimization of $\eta$ in L-ALFA is not reasonable.]
>
> In general, our design philosophy of L-ALFA is adversarial training. It maximizes the training objective to craft feature perturbation and minimizes the training objective to optimize model weights and the learnable “hyperparameter” (e.g., $\eta$ in equation 5). Note that there is a constraint $\mathcal{P}$ ball of $\eta$, where the summation of all $\eta_i$ is equal to 1. This constraint prevents the minimization to hurt the effect of feature augmentation and avoids trivial solutions (i.e., $\eta$=0). Moreover, as mentioned in the answer to Weakness 2, we adopt unbound perturbation generation and batch normalization in all networks. In addition, as shown in Table 6, we conduct five independent runs with different random seeds, and the performance is very stable, i.e., 94.65% (+- 0.08%).
>
> [Weakness 4: IJCNN 2019 paper.]
>
> Thanks for pointing out the literature. Our work is different from this IJCNN paper in three aspects: motivation, methods, and experiment design.
>
> (1) Motivation. IJCNN crafts strong adversarial perturbations to fool the model (“the perturbation is designed to be adversarial, which means that they are designed to significantly change the output of the network.”). Our method aims to generate moderate adversarially perturbed features to improve the generalization rather than to fool the model.
>
> (2) Methods. IJCNN injects adversarial perturbations in a randomly selected layer by virtual adversarial training and ad-hoc coefficient layers. We utilize PGD-AT to craft gradient-based perturbations, and further design the learnable L-ALFA framework to automatically learn the location and strength of generated feature perturbations.
>
> (3) Experiments. IJCNN only focuses on small-scale datasets (MNIST, NORB, CIFAR-10) with networks no more than eight layers. Our experiments are conducted on diverse datasets from small-scale CIFAR-10 and CIFAR-100 to large-scale ImageNet, with much larger network architectures than IJCNN, including ResNet-20s/56s, ResNet-18/-50/-101/-152.
>
> We hope that you can raise your score if you find our answers that address your questions. Thank you!

---

> > ### Comment · AnonReviewer1 · 2020-11-18
> > **Thank you for your response**
> >
> > Thank you for your response.
> > The responses to W2 and W4 make sense to me, though the manuscript need to be revised accordingly.
> >
> > [About W1]
> > As far as I understand, the efficiency of L-ALFA does not so much depend on the number of perturbed layers but heavily depends on choice of which layer to perturb. If you choose the layer near the input layer to perturb, computing gradient of L_at wrt delta requires backpropagation through almost all layers, which should substantially increase the computational cost due to multi-step PGD.
> >
> > [About W3]
> > I had already understood that "design philosophy of L-ALFA is adversarial training." However, considering this philosophy, it seems natural for me to maximize the objective function wrt eta. Why did the authors minimize it?

---

> > > ### Author Response · Authors · 2020-11-19
> > > **Response to Reviewer #1 [More About W1 & W3]**
> > >
> > > [More for W1: Efficiency Cost]
> > >
> > > Yes, you are right. The efficiency of L-ALFA not only depends on the number of perturbed layers but also depends on the choice of which layer to perturb. And perturbing the input layer costs more than perturbing the last layer.
> > >
> > > Fortunately, from the extensive and systematical study of layer preference of ALFA (Figures 2 and 3), applying the adversarial feature augmentation to the last layer benefits more to the generalization ability. As shown in Figure 4, L-ALFA also automatically learns the effective strategy of only perturbing the last block feature embeddings (i.e., only assign a large coefficient to the last layer perturbations), which is consistent with our observations made from ALFA.
> > >
> > > Therefore, a possible more efficient implementation of L-ALFA is only considering the last few layers since they are the top-performing candidates.
> > >
> > > [More for W3: Maximize or Minimize?]
> > >
> > > Thanks for your insightful comments. We think it mainly depends on the goal. For example, if the goal is to learn an effective adversarial perturbation to fool the model (i.e., robustness), we need to maximize the objective function wrt eta. If the goal is to find better configurations to improve clean accuracy (i.e., generalization ability), we need to minimize the $\eta$ so that the found $\eta$ is more beneficial for generalization improvements.
> > >
> > > To further address your concern, we also implemented the maximization version and provided the results here: Baseline (94.30%) v.s. L-ALFA w. min $\eta$ (94.62%) v.s. L-ALFA w. max $\eta$ (94.32%). The experiments are conducted on CIFAR-10 with ResNet-18.
> > >
> > > Please let us know if you have any additional concerns. And we are happy to further address them. Thank you.

---

> > > > ### Comment · AnonReviewer1 · 2020-11-23
> > > > **Thank you for your response**
> > > >
> > > > If choosing the last layer always works best, we do not need to use L-ALFA but just use ALFA at the last layer. However, since the difference of ALFA from prior works is marginal, it substantially reduces the novelty of this work. This is why I am focusing on the advantage and validity of L-ALFA in this review.
> > > >
> > > > About computational efficiency, yes, the proposed method can be efficient, if we appropriately choose candidates of layers to perturb. However, since the aim of L-ALFA is to automatically choose which layer to perturb, it sounds like a chicken-and-egg problem.
> > > >
> > > > Although it is clear that maximizing loss wrt \eta leads to improving robustness, it is not yet clear to me why minimizing loss wrt \eta improves the generalization ability. Could you provide intuitive explanation on this point? I really thank the authors to provide an additional experimental result on this point, but I am still struggling to understand why such a result is obtained.

---

> > > > > ### Author Response · Authors · 2020-11-23
> > > > > **Response to Reviewer #1 [Feedbacks 1-3]**
> > > > >
> > > > > Thanks for your constructive feedback. We truly value your acknowledgment of our efforts in the extra experiments. Below, we provide new responses to your concerns.
> > > > >
> > > > > **[Feedback 1: The Advantage and Validity of L-ALFA]**
> > > > >
> > > > > Although applying ALFA to the last layer is an effective way, L-ALFA remains its necessities and validities from the following three perspectives:
> > > > >
> > > > > - Although there exists a consistent observation that applying ALFA to the last layer is the top-performing setting across multiple datasets and diverse backbone networks, it does not mean this conclusion will hold for any unseen dataset and networks. As shown in Figure 2, the achieved performance of applying ALFA to Block2 (i.e., an intermediate layer) and Block3 (i.e., the last layer)  are almost the same. In contrast, L-ALFA is capable of automatically learning a satisfying configuration for adversarial feature augmentations under new scenarios.
> > > > >
> > > > > - It is worth mentioning that even if we adopt the “last layer” prior, it still needs arduous tuning for an appropriate attack strength for ALFA. Our proposed L-ALFA eliminates laborious tuning by automatically adjusting the perturbation magnitude of each perturbed feature. L-ALFA is much more user friendly for beginners who have little experience of hyperparameter tuning, in the machine learning community.
> > > > >
> > > > > - As presented in Tables 3 and 7, L-ALFA moderately surpasses ALFA on CIFAR-100 with ResNet-56s, which demonstrates there exist better-performing layer combinations (e.g., 0.42\*Block9 + 0.15\*Block8 + 0.14\*Block7 + ...) for introducing feature perturbations compared to the single last layer option.
> > > > >
> > > > > **[Feedback 2: Computational Efficiency]**
> > > > >
> > > > > Thanks for the comments. The aim of L-ALFA is to automatically (1) choose which layer to perturb and (2) control the perturbation strength of adversarial augmentations. For the second point, as stated in Feedback 2, L-ALFA can always stay efficient and effective, eliminating laborious tuning by automatically adjusting the perturbation magnitude. As for the first point of layer picking, we would like to say L-ALFA provides a trade-off between efficiency and effectiveness. On the one hand, if L-ALFA is allowed to select from a large pool of candidate layers (e.g., dimension($\eta$)=9), it is potentially capable of learning better-performing layer combinations with higher cost. On the other hand, performing L-ALFA to the shrank pool of candidate layers (e.g., considering the last few layers) enjoys an efficiency bonus while possibly missing some advanced configurations for perturbed layers.
> > > > >
> > > > >
> > > > > **[Feedback 3: Intuition of Minimize $\eta$]**
> > > > >
> > > > > We sincerely appreciate your acknowledgment of our effects on extra experiments. We try our best to provides intuitive explanations from the following two perspectives:
> > > > >
> > > > > - *Treating learning $\eta$ as a special “normalization”.* After generating the adversarial feature perturbations, the learnable $\eta$ serves as a special “normalization” to control the magnitude of crafted perturbations. The goal of such “normalization” is to find a better generalizable local minima in the loss landscape, and therefore the learning of $\eta$ needs to follow the gradient descent direction and decreasing the training loss (i.e., min $\eta$).
> > > > >
> > > > > - *Treating $\eta$ as model parameters.* Adversarial feature augmentations play a similar role as implicit data augmentation. Specifically, utilizing raw images together with generated adversarial features as input samples, to train the classification models by optimizing all model parameters (includes $\eta$). In this sense, it is quite natural to solve the optimization by minimizing the training loss (i.e., min $\eta$).
> > > > >
> > > > > Overall, as mentioned by Reviewer #1, the results of our extra experiment are consistent with our design intuitions. (i.e., Baseline (94.30%) v.s. L-ALFA w. min $\eta$ (94.62%) v.s. L-ALFA w. max $\eta$ (94.32%). The experiments are conducted on CIFAR-10 with ResNet-18.)
> > > > >
> > > > >
> > > > > Please don’t hesitate to let us know if you have any additional concerns. And we are happy to further address them. Thank you!

---

### Official Review · AnonReviewer2 · 2020-10-28
**This paper proposes the use of adversarial training at the feature level layers to improve generalization of Neural Networks. In particular,the authors propose to use PGD (Projected Gradient Descent) adversarial training at intermediate layers to increase the standard accuracy on clean images belonging to the test set.**

**Rating:** 4
**Confidence:** 5

**Review:**

Pros:

Method is clearly stated, and the learning adaptive perturbation strength seems novel.

Provided experimental results for large datasets like ImageNet.

Major Concerns:

The method proposes crafting adversarial perturbation at different layers of the network. A small change in features space may correspond to a large change in input space for neural networks, hence it is doubtful whether the perturbations crafted will be meaningful. The method which attempts to cause perturbations in the feature space also ensures that perturbed image is similar to original image
(imperceptibility constraint) [3].

The authors mention use of unbounded perturbation size in section 4.2 which goes against the goal of adversarial training (i.e. same prediction in the neighborhood of the image). Also, if the perturbation is unbounded then there is no requirement of Projection step in PGD (Projection is done for enforcing the boundedness of perturbation). Could the authors please clarify this?

The intuition why this method works is also not too clear, as it has been shown in [1] that adversarial training leads to drop in standard accuracy. But in this paper adversarial training has been shown to increase standard accuracy even without using any adaptive parameters (like batch normalization layers used in AdvProp [2]). This simply goes against the established facts.

It is hard to distill what settings work well across different datasets. As seen in Table 5, for CIFAR-10, changing from 1 step PGD to 5 step PGD increases accuracy, whereas it decreases in case of ImageNet. Also there exist only a certain range of step size values for which standard accuracy increases. Could the authors kindly offer additional theoretical or intuitive explanations to clarify the same? This would
help substantially improve the submission.

Other questions:

The authors could  provide info on fooling achieved by crafted adversaries which is required for confirming the success of adversary creation. Also, the adversarial training method proposed in this paper shows minimal increase in adversarial robustness which also supplements the need for details on fooling rate. Typically adversarial training  increases its robustness [5], which is not observed with the proposed method. Could the authors clarify this?

The accuracy reported for the ImageNet models is lower than Torchvision models [4] as the authors use the same code as PyTorch repo it is unexpected. This may be due to the use of 10% training data for validation which is not required in ImageNet, as it already has a validation set. (This is important as the magnitude of improvements are not very large over the standard accuracy in case of ImageNet and is
even smaller when compared with Torchvision models). The Torchvision ResNet 50 model obtains 76.15% accuracy which is comparable to ALFA (76.23%) (present in Table 3).


As the perturbations are calculated for each layer, the complexity of the method would increase with large networks like ResNet-152. So it is unclear if the proposed method will continue to be cost efficient in comparison to pixel-based methods.

Also, could the authors clarify the network architecture used for comparison of the time complexity in Section 4.3 for ALPHA and AdvProp [2]?

Overall, this seems a very complex method without any theoretical grounding to increase the generalization performance.

[1] Zhang, H., Yu, Y., Jiao, J., Xing, E., Ghaoui, L.E. & Jordan, M.. (2019). Theoretically Principled Trade-off between Robustness and Accuracy. Proceedings of the 36th International Conference on Machine Learning, in PMLR 97:7472-7482

[2]Xie, C., Tan, M., Gong, B., Wang, J., Yuille, A. L., & Le, Q. V. (2020). Adversarial examples improve image recognition. In Proceedings of the IEEE/CVF Conference on Computer Vision and Pattern Recognition (pp. 819-828).

[3]Ganeshan, A., & Babu, R. V. (2019). FDA: Feature disruptive attack. In Proceedings of the IEEE International Conference on Computer Vision (pp. 8069-8079).

[4]https://pytorch.org/docs/stable/torchvision/models.html

[5] Madry, A., Makelov, A., Schmidt, L., Tsipras, D., & Vladu, A. (2018, February). Towards Deep Learning Models Resistant to Adversarial Attacks. In International Conference on Learning Representations.

---

> ### Author Response · Authors · 2020-11-14
> **Response to Reviewer #2 [Major Concerns 1-4]**
>
> Thanks for your insightful comments. Below, we provide detailed responses to your concerns.
>
> [Major Concern 1:  It is doubtful whether the perturbations crafted will be meaningful.]
>
> Our work aims to enhance image recognition on the clean test set; therefore, we do not have the constraint that the perturbed image needs to be similar to the original image (i.e., the imperceptibility constraint).  More specifically, our method performs “adversarial data augmentation”, with a focus on leveraging adversarial perturbations in the feature space for regularization, rather than generating actual adversarially perturbed images. We will make this clear in the revision.
>
> [Major Concern 2: Unbounded perturbation.]
>
> Our generated perturbations are applied to intermediate feature embeddings. Since we only care about the final clean accuracy, there is no constraint that we need to craft adversarial perturbations in the neighborhood of the image. For the unbounded perturbation generation, we do not adopt the projection process. In practice, the magnitude of crafted feature perturbation steadily stays in a range from 0.97 to 1.10 under the $\ell_2$ norm. We will make this clear in the revision.
>
>
> [Major Concern 3: Clarify the intuition.]
>
> Our method is not against the facts in [1]. TRADES [1] found that vanilla adversarial training on the input pixel space leads to standard accuracy degradation. AdvProp [2] utilizes adversarial examples with an auxiliary batch normalization to improve standard accuracy.  Our work is different in that we introduce adversarial perturbation to feature embeddings rather than raw pixels, which improves the standard accuracy of image recognition. Our method is also well-motivated since adversarial feature augmentation has been widely proved beneficial for model generalization, such as [3] in computer vision, [4,5] in natural language processing, and [6] in vision+language tasks.
>
> [Major Concern 4: Insights of hyperparameter selection.]
>
> As mentioned by R2, to obtain standard accuracy improvement, there is a safe zone of hyperparameter selection for perturbation strength. The immoderate weak perturbation yields marginal effects, while excessive strong perturbations may lead to over-smoothed decision boundaries and incur performance degradation. Thus, learning a proper perturbation strength is important for our proposed ALFA. Moreover, in order to avoid laborious tuning across diverse datasets and backbones, we design learnable adversarial feature augmentation (L-ALFA) to automatically learn the best hyperparameter configurations.
>
> [1] Theoretically Principled Trade-off between Robustness and Accuracy, ICML 2019.
>
> [2]  Adversarial examples improve image recognition, CVPR 2020.
>
> [3] Improved Sample Complexities for Deep Networks and Robust Classification via an All-Layer Margin, ICLR 2020.
>
> [4] FreeLB: Enhanced Adversarial Training for Natural Language Understanding, ICLR 2020.
>
> [5] SMART: Robust and Efficient Fine-Tuning for Pre-trained Natural Language Models through Principled Regularized Optimization, ACL 2020.
>
> [6] Large-Scale Adversarial Training for Vision-and-Language Representation Learning, NeurIPS 2020.

---

> > ### Comment · AnonReviewer2 · 2020-11-24
> > **Thanks for your response**
> >
> > I thank the authors for providing answers in detail and appreciate them for running additional experiments. Although some of the things still don’t seem apt to me as per my understanding. I have mentioned them below:
> >
> > Major Concern 1: To my understanding, any augmentation mustn't differ from the original sample semantically. As your perturbation is small as mentioned in reply to major concern 2, I think this might be happening in the majority of the cases, But still, I am not sure whether it is guaranteed to happen in every case.
> >
> > Major Concern 2: As you aren’t doing any projection why the paper is mentioning PGD in formulas and in algorithms. Also in what space the difference in l_2 norm of perturbation is calculated? If it is calculated in feature space, I still don’t understand what meaning it will have in input space. Can authors please clarify this?
> >
> > Major Concern 3: As authors have referenced paper [3], which also does adversarial perturbation for all feature layers and obtains an increase in generalization performance. Can authors please explain why [3] is not a good baseline for comparison with ALFA?.
> >
> > Major concern 4: Yes, I do understand L-ALFA is taking away the laborious tuning. But the results provided with L-ALFA are very limited. Also, I share a similar concern as Reviewer 1,  that why $\eta$ should minimize the loss, since adversaries are usually chosen by maximizing the loss. Could authors please shed some light on this phenomenon?

---

> > > ### Author Response · Authors · 2020-11-24
> > > **Response to Reviewer #2 [More about Major Concerns 1-3]**
> > >
> > > [More for Major Concern 1: Semantic Preserving Augmentations]
> > >
> > > Generally, it is true that “any augmentation mustn't differ from the original sample semantically”. However, it does not mean any augmentation must have explicit constraints to limit the distance to original samples. For example, the common choice for deciding the strength of traditional data augmentation like color jittering and gaussian blur, is handcrafted tuning rather than applying explicit constraints [6], and our adversarial feature augmentations (ALFA) also follow these routines. For learnable data augmentation [4,5], the strength of augmentation is mainly automatically learned, and our L-ALFA has the same fashion. To further address your concerns, we evaluate all generated adversarial features to see whether it will flip the prediction (i.e., change semantic meanings). We find that 0% of adversarial features generated by ALFA with ResNet-20s on CIFAR-10 flip the prediction. It comes as no surprise since our aim is to utilize moderate adversarial features to enhance generalization ability rather than to fool the models.
> > >
> > >
> > > [More for Major Concern 2: Formulation and Perturbations]
> > >
> > > Thanks for the suggestions. Our approaches are derived from adversarial training. For completeness of the derivation, we follow the same notation and terms in the adversarial training. We have added sentences to clarify this confusion in Section 3.2 of our updated paper.
> > >
> > > The difference in $\ell_2$ norm of perturbations is calculated in the intermediate feature spaces. It is worth mentioning that our method performs “adversarial data augmentation”, with a focus on leveraging adversarial perturbations in the feature space for regularization, rather than generating actual adversarially perturbed images. In other words, the crafted feature perturbations would not affect the input space. Specifically, if we want to generate the adversarial features for the last layer, the network only backpropagates to the last layer, and therefore the generated perturbations do not have an influence on the input space.
> > >
> > >
> > > [More for Major Concern 3: Comparisons with [3]]
> > >
> > > Thanks for the suggestions. [3] mainly focus on theoretical analyses for adversarial feature augmentations, while they conduct empirical investigations only on the small-scale datasets (i.e., CIFAR-10 and CIFAR-100) with WRN-16-10 and WRN-28-10. As shown in Figures 2 and 3, we indeed consider a similar baseline that applies adversarial feature perturbations to all feature layers. We notice that injecting feature perturbations to the last layer achieves superior performance to the setting of perturbing all layers. To further address your concerns, we will include a rigorous experimental comparison with [3] in the final version since there are only a few hours left for the rebuttal period.
> > >
> > > [3] Improved Sample Complexities for Deep Networks and Robust Classification via an All-Layer Margin, ICLR 2020.
> > >
> > > [4] AutoAugment: Learning Augmentation Strategies from Data
> > >
> > > [5] Learning Data Augmentation Strategies for Object Detection
> > >
> > > [6] A Simple Framework for Contrastive Learning of Visual Representations

---

> > > ### Author Response · Authors · 2020-11-24
> > > **(Continued) Response to Reviewer #2 [More about Major Concerns 4]**
> > >
> > > [More for Major Concern 4: Validity of L-ALFA and Intuitions of Minimizing $\eta$]
> > >
> > > *I. The Validity of L-ALFA.*
> > >
> > > We respectfully disagree “the results provided with L-ALFA are very limited”.
> > >
> > > - We observe from Table 2, 3, and 7, (ALFA v.s. L-ALFA) consistently improves the generalization of trained networks by (0.32% v.s. 0.35%) on CIFAR-10, (1.18% v.s. 1.14%) on CIFAR-100, and (0.91% v.s. 1.02%) on ImageNet
> > > Note that here ALFA is carefully tuned yet L-ALFA are automatically learned. The main goal of L-ALFA is to eliminate laborious tuning of key parameters such as locations and strength of feature augmentations and meanwhile, achieves comparable performance improvements to ALFA. Besides, L-ALFA is much more user friendly for beginners who have little experience of hyperparameter tuning, in the machine learning community.
> > >
> > > - L-ALFA is capable of automatically learning a satisfying configuration for adversarial feature augmentations under new scenarios. As presented in Tables 3 and 7, L-ALFA moderately surpasses ALFA on CIFAR-100 with ResNet-56s, which demonstrates there exist better-performing layer combinations (e.g., 0.42\*Block9 + 0.15\*Block8 + 0.14\*Block7 + ...) for introducing feature perturbations compared to the single last layer option.
> > >
> > > *II. The Intuitions of Minimizing $\eta$*
> > >
> > > We respectfully point out that $\eta$ is not designed for generating adversaries, but for learning better feature adversaries’ combinations to enhance generalization ability. As shown in Equation 5, adversaries $\delta^{(i)}$ are still generated by maximizing the loss and then optimizing $\eta$ together with model weights to improve the generalization by minimizing the loss. Specifically, we further provide intuitive explanations from the following two perspectives:
> > >
> > > - *Treating learning $\eta$ as a special “normalization”.* After generating the adversarial feature perturbations, the learnable $\eta$ serves as a special “normalization” to control the magnitude of crafted perturbations. The goal of such “normalization” is to find a better generalizable local minima in the loss landscape, and therefore the learning of $\eta$ needs to follow the gradient descent direction and decrease the training loss (i.e., min $\eta$).
> > >
> > > - *Treating $\eta$ as model parameters.* Adversarial feature augmentations play a similar role as implicit data augmentation. Specifically, utilizing raw images together with generated adversarial features as input samples, to train the classification models by optimizing all model parameters (includes $\eta$). In this sense, it is quite natural to solve the optimization by minimizing the training loss (i.e., min $\eta$).
> > >
> > > Overall, as mentioned by Reviewer #1’s feedback, the results of our extra experiment are consistent with our design intuitions. (i.e., Baseline (94.30%) v.s. L-ALFA w. min $\eta$ (94.62%) v.s. L-ALFA w. max $\eta$ (94.32%). The experiments are conducted on CIFAR-10 with ResNet-18.
> > >
> > > Please let us know if you have any additional concerns. And we are happy to further address them. Thank you.
> > >
> > > [3] Improved Sample Complexities for Deep Networks and Robust Classification via an All-Layer Margin, ICLR 2020.
> > >
> > > [4] AutoAugment: Learning Augmentation Strategies from Data
> > >
> > > [5] Learning Data Augmentation Strategies for Object Detection
> > >
> > > [6] A Simple Framework for Contrastive Learning of Visual Representations

---

> ### Author Response · Authors · 2020-11-14
> **(Continued) Response to Reviewer #2 [Other Questions 1-5 & Summary]**
>
> [Other Question 1: The success of adversary creation.]
>
> The crafted adversarial features serve as “implicit data augmentation” for model regularization rather than aiming at fooling the model. As mentioned in the answer to ‘Major Concern 3’, our goal is utilizing adversarially augmented features to improve the standard accuracy of image recognition. The moderate robustness boost is an extra bonus.
>
> [Other Question 2: Validation datasets and Torchvision model performance.]
>
> The evaluation of ImageNet models is conducted on the official validation set of ImageNet [2] since there is no test set in ImageNet. This is why we split the training dataset to form our own val set for early stopping. To address your concern, we also conduct experiments with a full (100%) ImageNet training set, Baseline (75.98%) vs. ALFA (76.51%). In addition, we also provide the EfficientNet-B0 results on ImageNet here, Baseline (77.04%) vs. ALFA (77.48%). Extensive experiments demonstrate that ALFA achieves consistent generalization improvement over multiple backbones and diverse datasets.
>
> [Other Questions 3 and 4: Cost efficiency.]
>
> The comparison of time complexity in Section 4.3 is conducted on ResNet-18 architecture. For ALFA, we only apply adversarial perturbations to a single layer, which allows ALFA to maintain cost efficiency even for deep networks. The efficiency of L-ALFA (compared to pixel-based methods) mainly depends on the number of perturbed layers (i.e., the dimension of $\eta$ in equation 5).  We agree that if there are hundreds of layers that need to be perturbed, L-ALFA may not be efficient enough. Fortunately, as shown in Table 8, a moderate number of perturbed layers (e.g. dim($\eta$)=9) is enough to demonstrate the effectiveness of  L-ALFA. We provide an updated time comparison with ResNet-18 per training epoch here: AdvProp (123s) vs. ALFA (28s) vs. L-ALFA with dim($\eta$)=9 (91s) vs. Standard training (23s).
>
> [Other Question 5: Theoretical grounding.]
>
> Theoretical guarantees and analyses are so far still missing in the field, as most adversarial training papers are empirical. We provide some insights about where the performance improvements come from. Similar to the discussion in the NLP field [5,6], we believe adversarial feature augmentation works as an implicit regularization for training, which smoothes the decision boundary [5] and alleviates overfitting [5], therefore obtaining generalization improvements.
>
> [Summary]
>
> As a summary, our method aims at improving the generalization performance via feature augmentation instead of focusing on adversarial robustness, and there is no imperceptibility constraint.
>
> We believe the reviewer has some misunderstanding about the key idea in our method, and we are sorry for this confusion. We hope that you can raise your score if you find our answers that address your questions. Thank you!
>
> [1] Theoretically Principled Trade-off between Robustness and Accuracy, ICML 2019.
>
> [2]  Adversarial examples improve image recognition, CVPR 2020.
>
> [3] Improved Sample Complexities for Deep Networks and Robust Classification via an All-Layer Margin, ICLR 2020.
>
> [4] FreeLB: Enhanced Adversarial Training for Natural Language Understanding, ICLR 2020.
>
> [5] SMART: Robust and Efficient Fine-Tuning for Pre-trained Natural Language Models through Principled Regularized Optimization, ACL 2020.
>
> [6] Large-Scale Adversarial Training for Vision-and-Language Representation Learning, NeurIPS 2020.

---

### Official Review · AnonReviewer4 · 2020-10-28

**Rating:** 6
**Confidence:** 4

**Review:**

In this paper, the authors suggest a method for adversarial feature-level augmentation, mainly framed as an approach to improve the clean-set accuracy rather than adversarial example robustness. The authors also propose a learnable version (LALFA), to automatically learn the location and strength of the perturbations.
Overall, I think this is an interesting paper that can be considered for publication at ICLR. The following elaborates it further:

Strengths:
* Even though the adversarial feature augmentations at the feature level are not unprecedented, but the learnable tuning of the location/strength is an interesting approach that can potentially avoid expensive hyperparameter optimization.
* The shown improvements of the best-achieved models seem consistent over the baselines, across different datasets and models.

Weaknesses:
* The obtained improvements are moderate on smaller networks and become marginal with deeper counterparts.
* The performance and the offered advantages seem quite sensitive to the choice of hyperparameters in ALFA (Tables 4 and 5). I wonder, at least, how stable/conclusive the comparisons are when transferring the selected model from one set to another, e.g. validation to test.

Further detailed comments:
* " L-ALFA saves toilsome tuning by automatically adjusting the strength and locations of adversarial feature augmentations" => It is fair to note that this is achieved at the expense of introducing a new hyper-parameter to tune, namely \alpha.
* An ablation study on the L1 regularization could have been useful.
* In Table 1 and some of the other numerical comparisons, except for Table 6, where standard deviations are reported: I wonder how significant are these comparisons? Are the differences meaningful when considering the intra-experiment variations?
* Insights on why MoEx, one of the three baselines compared against, is performing worse than both random noise and normal training, would be helpful.
* Equation (1) defines \delta \in B_\epsilon(x) and uses it as an offset to x: f(x+\delta;\theta). Later B_\epsilon(x) is defined as "The norm ball B_\epsilon is centered at x with radius \epsilon": These are not consistent, if \delta is used as an offset, it cannot be sampled around x.
* \mathcal{L}_{at} has been used inconsistency across equation (1) and equations (3) and (4), on the first argument; f_i(x;\theta^{(i)}+\delta^{(i)}) in equation (3) and (4) misses the second part of the network, transforming the intermediate feature maps to the required predictions.
* \mathcal{L}_{at} is used in equation 1, but is elaborated after equation 2.
* Defining both F(x;\theta) and f(x;\theta) is referring to the neural network, and its output seems unnecessary and confusing; besides, F(x;\theta) is never referenced before.
* f(x+\delta,\theta) => f(x+\delta;\theta)
* Adding a row in table 7, representing the results from ALFA, will make the direct comparison between ALFA and L-ALFA easier.
* "ResNet-18 has ... and twenty convolutional layers" => It has seventeen convolutional layers, I think. Please clarify on dim(\eta) = 20 in Table 8. Are some of the pooling layers outputs also taken?
* typo: "a unduly"

---

> ### Author Response · Authors · 2020-11-19
> **Response to Reviewer #4 [Weakness 1&2]**
>
> Thanks for your insightful comments. Below, we provide detailed responses to your concerns.
>
> [Weakness 1: Moderate and Marginal Improvements.]
>
> We respectfully do not agree that the improvements are marginal. Our proposed methods ALFA, as an implicit data augmentation or regularization, achieves consistent generalization improvements over strong baselines on CIFAR-10, CIFAR-100, and ImageNet benchmarks across different backbone networks for image recognition. Note that, these datasets are highly competitive, and it is impressive to obtain further improvements based on baselines’ almost saturated performance.
>
> [Weakness 2: The Stability of Hyperparameters from Validation to Test.]
>
> As mentioned in Section 4.1, the original training dataset is randomly split into 90% training and 10% validation in our experiments. The early stopping technique is applied to find the top-performing checkpoints on the validation set. Then, the selected checkpoints are evaluated on the test set and report the performance. From our observations, the hyperparameters are quite stable from validation to test sets. To further address your concerns, we provide the code as the additional supplementary material that will be uploaded before the end of the rebuttal period.

---

> ### Author Response · Authors · 2020-11-19
> **(Continued) Response to Reviewer #4 [Detailed Comments 1-12]**
>
> [Detailed Comment 1: New Hyperparameters in L-ALFA.]
>
> Yes, we agree that L-ALFA introduces a new hyperparameter, $\gamma$, to achieve the automatic adjustment of the strength and locations of adversarial feature augmentation. In our experiments, $\gamma$ can be roughly chosen from {0.5, 1.0, 2.0} and obtain similar performance improvements. A detailed ablation is provided in the response to Detailed Comment 2.
>
> [Detailed Comment 2: Ablation Studies on the $\ell_1$ regularization.]
>
> Ablation results of ResNet-56s on CIFAR-100 are collected in Table S1. We observe that $\gamma$ can be roughly chosen from {0.5, 1.0, 2.0} and obtain similar competitive performance. Excessive large or small $\gamma$ will incur performance degradation.
>
> Table S1. Ablations of L-ALFA on the $\ell_1$ regularization of $\gamma$. Note that the baseline performance is 71.22%.
>
> |$\gamma$|0.1|0.2|0.5|1.0|2.0|10.0|
> |:-:|:-:|:-:|:-:|:-:|:-:|:-:|
> |Accuracy|70.68%|72.13%|72.35%|72.40%|72.38%|71.92%|
>
> [Detailed Comment 3: Standard Deviation Errorbar.]
>
> Due to the limited time, here we provide the errorbar for ResNet-18 on ImageNet, ResNet-56s on CIFAR-10, and CIFAR-100. The mean and standard deviations are reported in Table S2. Each number is calculated with five independent runs. We observe that the obtained improvements of ALFA are statistically significant (i.e., highly non-overlapped errorbar).
>
> Table S2. Repeat experiments on each dataset.
>
> |Settings|CIFAR-10|CIFAR-100|ImageNet|
> |:-:|:-:|:-:|:-:|
> |Baseline|93.03% $\pm$ 0.10%|71.22% $\pm$ 0.08%|69.38%  $\pm$ 0.03%|
> |ALFA|94.72% $\pm$ 0.12%|72.36%  $\pm$ 0.13%|70.19%  $\pm$ 0.09%|
>
> [Detailed Comment 4: Insights on MoEx.]
>
> It is fair to note that this is not our main competitive baseline. In experiments of MoEx, we have tried the default hyperparameters in the original paper and other carefully tuned configurations for better performance. However, the results are not satisfactory. We have communicated with the authors in the MoEx paper, and cannot achieve meaningful conclusions yet. We will continue to investigate it and try our best to provide analysis in the future.
>
> [Detailed Comments 5-10, 12: Typos and Confusing Notations.]
>
> Thank you so much for such detailed and helpful suggestions. We will carefully correct all the typos and confusing notations according to your suggestions in our modified draft, before the end of the rebuttal period.
>
> [Detailed Comment 11: Clarify on $\mathrm{dim}(\eta)$=20.]
>
> Thanks for the detailed comments. Yes, there are seventeen convolutional layers in the main structure of ResNet-18. In addition, in the skip connection of residual blocks 2, 3, and 4, each one has a convolutional layer. Thus, 17+3=20 positions for injecting adversarial feature perturbations.

---

> > ### Comment · AnonReviewer4 · 2020-11-23
> >
> > I would like to thank the authors for providing detailed responses, to my raised questions and concerns.
> > I am satisfied with the answers provided for W1, W2, DC4, DC5,7-12, and the clarifications added, and the corrections made to the paper.
> >
> > * DC1&2: I appreciate that the authors have conducted and sensitivity analysis on the $\gamma$ hyperparameter. However, it is worth mentioning that it doesn't guarantee that the same hyperparameter values would hold optimal to any other datasets and therefore this hyperparameter remains to be optimized for new datasets, over the (more costly) L-ALFA's performance.
> >
> > * DC3: I thank the authors for providing the error bars for a number of comparisons and I agree that the improvements in those cases look significant. However, the set of comparisons, are rather on pairs with larger margins between the proposed method and the baseline, i.e. on smaller networks. Showing the significance of improvements on the larger models where the differences are less considerable, could have better helped the authors convey their message, I think.
> >
> > * DC6: The inconsistent notation for using $\mathcal{L}_{at}$ between equations (1), (3), and (4), seems to be remaining in the revised manuscript. But I also understand that making the notation more accurate and consistent could also result in complications. So I think in this case, this could be compromised for simplicity.

---

### Author Response · Authors · 2020-11-23
**General Response**

Dear Reviewers:

We thank all reviewers for their constructive comments and insightful suggestions to strengthen this work. We are glad that R3 and R4 recognized our paper as novel and experimental results convincing.

In addition to the specific response below, here we summarize our updates:

- We have carefully corrected all the typos and confusing notations, and enriched the related works in our revisions.

- As mentioned by reviewer #1-4, plenty of new experimental results are provides, including ablation studies of $\ell_1$ regularization in L-ALFA, more running time analyses of ALFA and L-ALFA, and more comparisons between AdvProp and ALFA on multiple datasets with diverse networks backbones (e.g., EfficientNet-B0).

- As for reproductivity, we have released our codes and pre-trained models as additional supplementary material.

Please don’t hesitate to let us know of any additional comments on the paper or on the responses. We thank all reviewers’ time again.

---

### Decision · Program_Chairs · 2021-01-07
**Final Decision**

**Decision:**

Reject

**Comment:**

Adversarial training is usually done on the image space by directly optimizing the pixels. This paper suggests the adversarial training over intermediate feature spaces in the neural network. The idea is very simple. The authors have done extensive experiments to justify its performance. But the performance gain though this idea seems to be marginal. Further, the layer to conduct the adversarial training can be optimized within the framework, which aligns with the general autoML idea. The new version L-ALFA has been well introduced, but unfortunately, the practical result can be very straightforward, that is just to select the final layer. The more important ALFA hyperparameters that would most benefit from automatic tuning are not sufficiently treated.  There have been extensive discussions between the authors and the reviewers. After incorporating the reviewers' comments, the paper will have a good chance to be accepted at another venue.